# Molecular Functionality of Cytochrome P450 4 (CYP4) Genetic Polymorphisms and Their Clinical Implications

**DOI:** 10.3390/ijms20174274

**Published:** 2019-08-31

**Authors:** Yazun Bashir Jarrar, Su-Jun Lee

**Affiliations:** 1Department of Pharmacy, College of Pharmacy, Alzaytoonah University of Jordan, 11734 Amman, Jordan; 2Department of Pharmacology and Pharmacogenomics Research Center, Inje University College of Medicine, Inje University, Busan 47392, Korea

**Keywords:** *CYP4* genes, genetic polymorphisms, 20-HETE, fatty acid, arachidonic acid, SNPs, molecular functionality, metabolism, lamellar ichthyosis, Bietti’s crystalline dystrophy

## Abstract

Enzymes in the cytochrome P450 4 (CYP4) family are involved in the metabolism of fatty acids, xenobiotics, therapeutic drugs, and signaling molecules, including eicosanoids, leukotrienes, and prostanoids. As CYP4 enzymes play a role in the maintenance of fatty acids and fatty-acid-derived bioactive molecules within a normal range, they have been implicated in various biological functions, including inflammation, skin barrier, eye function, cardiovascular health, and cancer. Numerous studies have indicated that genetic variants of *CYP4* genes cause inter-individual variations in metabolism and disease susceptibility. Genetic variants of *CYP4A11*, *4F2* genes are associated with cardiovascular diseases. Mutations of *CYP4B1*, *CYP4Z1*, and other *CYP4* genes that generate 20-HETE are a potential risk for cancer. *CYP4V2* gene variants are associated with ocular disease, while those of *CYP4F22* are linked to skin disease and *CYP4F3B* is associated with the inflammatory response. The present study comprehensively collected research to provide an updated view of the molecular functionality of *CYP4* genes and their associations with human diseases. Functional analysis of *CYP4* genes with clinical implications is necessary to understand inter-individual variations in disease susceptibility and for the development of alternative treatment strategies.

## 1. Introduction

Cytochrome P450s (CYPs) are a superfamily of enzymes located either in the inner membrane of mitochondria or in the endoplasmic reticulum membrane of eukaryotic cells. There are 57 CYP proteins encoded in the human genome, which are responsible for the metabolism of numerous endogenous and exogenous compounds [1,2,3]. CYPs mainly oxidize these compounds to generate more hydrophilic metabolites, enhancing their excretion outside the body and thus playing a major role in the detoxification of toxic chemicals [1]. Generally, CYP families 1, 2, and 3 include major xenobiotic-metabolizing enzymes responsible for their major roles in pharmacogenomics risk, while CYP4 enzymes are involved in the metabolism of fatty acids, with their close links to genetic disease risk. Fatty acid metabolism by CYP4 enzymes is responsible for the elimination of excess free fatty acids from the body, as well as for the synthesis of proper levels of bioactive fatty acid molecules [4]. The present review focused on the CYP4 family of enzymes in terms of their functional roles, genetic variations, and influences on human diseases.

## 2. Classification and Tissue Distribution of the CYP4 Family

Although there are more than 11 subfamilies of CYP4 in different species, only 6 subfamilies of *CYP4* genes have been reported in humans. The human CYP4 subfamilies are CYP4A, B, F, V, X, and Z [2]. Seven *CYP4F* isoforms, *CYP4F2, CYP4F3A, CYP4F3B, CYP4F8, CYP4F11, CYP4F12*, and *CYP4F22*, are clustered on chromosome 19 and are encoded by six genes. *CYP4A* isoforms include *CYP4A11* and *CYP4A22* on chromosome 1 [3], and the remaining *CYP4* subfamily genes are *CYP4B1*, *CYP4V2*, *CYP4X1*, and *CYP4Z1* [4]. The major sites of CYP4A11 expression are the liver and kidney [5]. However, Jarrar et al. found that CYP4A11 protein was also highly expressed in human platelets to a similar level as in the human liver [6]. CYP4A22 expression has been reported in the human liver at very low levels, with poor enzyme activity compared to that of CYP4A11 [7]. Among the seven *CYP4F* genes, *CYP4F2*, *CYP4F3B*, *CYP4F11*, and *CYP4F12* are mainly expressed in the liver and kidney [8,9,10,11,12]. However, their relative contributions to the total amount of CYP4 enzymes in tissues are difficult to determine, as the high structural homology of these four enzymes has hampered the production of specific antibodies for the detection of each enzyme. In addition to the lack of specific antibodies, genetic polymorphisms and differing profiles of up- and downregulation among CYP4 enzymes have further complicated determination of the intrinsic amount of each enzyme in tissues. Currently, mass spectrometry is used to detect target proteins through measurement of specific peptides of the target protein [13,14]. The total amount of CYP4F protein in human liver was estimated as 18–128 pmol/mg liver microsomal protein [15]. One of the most abundant P450s, CYP3A4, was estimated at 64 pmol/mg liver microsomal protein [13], indicating that the contribution of CYP4F to the total P450 level is large. CYP4F3A is expressed in neutrophils and plays a major role in inflammation [16]. CYP4F8 is expressed in the prostate and seminal vesicles [17]. CYP4F22 is expressed in human skin and plays a major role in formation of the skin lipid barrier [18]. CYP4V2 is widely expressed in the liver and ophthalmic tissues and CYP4V2 defect has been linked to ophthalmic diseases, such as Bietti’s crystalline dystrophy [19]. CYP4B1 is expressed mainly in the lung and bladder tissues, and in smaller amounts in the liver [20]. CYP4X1 is expressed in the brain and bronchial airways [21], while CYP4Z1 is expressed in mammary tissue; these proteins are also overexpressed in cancer compared to normal cells [22]. Expression levels of CYP4 proteins are summarized in Table 1.

## 3. Role of the CYP4 Family in the Metabolism of Endogenous Compounds

The CYP4 family plays a major role in the metabolism of fatty acids, in most cases through oxidation of fatty acids and subsequent catalysis in the mitochondria to produce cellular energy. CYP4B metabolizes short-chain fatty acids (approximately 7 to 10 carbon fatty acids) [20], while CYP4A and CYP4V metabolize intermediate-chain fatty acids (C10 to 16) [19] and CYP4F catalyzes long-chain fatty acids (C16 to 26), such as prostanoids [43]. Decreasing the expression levels of the CYP4 family was associated with accumulation of fats in tissues such as the liver [44]. Therefore, decreased levels of CYP4 family proteins reduce the capacity for fat removal from tissues. Jarrar et al. [44] found that non-steroidal anti-inflammatory drugs caused fatty livers in treated mice, which were associated with significant downregulation of mouse *cyp4a12* gene expression in liver tissues. CYP4F2, 4F3B, 4A11, and 4V2 were found to oxidize arachidonic acid through ω-hydroxylation to 20-hydroxyeicosatetraenoic acid (20-HETE) [6,9,45,46], which is a vasoconstrictor and activator of platelet aggregation [47]. Several studies have reported that CYP4F and CYP4A are overexpressed in cardiovascular diseases, wherein they are correlated with 20-HETE production [48,49,50]. In addition, doxorubicin-induced cardiotoxicity was associated with increased 20-HETE production due to increased mRNA expression of rat CYP4A and CYP4F enzymes [51]. CYP4A11 and 4V2 oxidize saturated fatty acids such as lauric acid [23,41,52]. In addition to the metabolism of arachidonic acid and omega-3 polyunsaturated fatty acids, CYP4F2 has been reported to ω-hydroxylate leukotriene (LTA) 4 [53]. CYP4F3A in white blood cells catalyzes the ω-hydroxylation of leukotriene B4 to 20-hydroxy leukotriene B4, which is an important regulatory step of the inflammatory response [54]. Instead of ω-hydroxylation, CYP4F8 has been reported to hydroxylate prostaglandin (PG) E2 at position 19 [17]. Although epoxyeicosatrienoic acids (EETs) are synthesized by the CYP2C subfamily [55], they can be further ω-hydroxylated by CYP4 enzymes to 20-hydroxyepoxyeicosatrienoic acids (HEETs) [56].

## 4. Role of the CYP4 Family in the Metabolism of Drugs

The roles of most CYP4 family proteins in the metabolism of drugs and xenobiotic compounds appear to be minor compared to those of CYP1, 2, and 3. However, CYP4F2 metabolizes the ester prodrug of gemcitabine and the antiparasitic pafuramidine [57]. In addition, CYP4A11 exhibited metabolism of the immune suppressant tacrolimus to an inactive form [58]. Although the turnover rates were low compared to those of CYP3A4, CYP4F11 exhibited catalytic activity towards commonly used drugs such as erythromycin, benzphetamine, and chlorpromazine [36,37,59]. CYP4F12 has been reported to slowly metabolize the antihistamine ebastine [60] and the antifungal terfenadine [61]. CYP4 enzymes are indirectly involved in drug metabolism and drug responses. For example, CYP4F2 and CYP4F11 are involved in the metabolism of vitamin K, facilitating vitamin K inactivation and elimination [29,62]. The amount of active vitamin K is important for maintenance of warfarin dosing, as it is metabolized strongly by CYP2C9 [63,64], indicating that CYP4 enzymes are indirectly involved in warfarin dose maintenance. CYP4 enzymes show catalytic activity toward various fatty acids and their metabolites have the potential to act as ligands or activators of nuclear receptors, such as peroxisome proliferator-activated receptors (PPARs) [65,66]. Therefore, drugs targeting the activation or inactivation of PPARs may show altered pharmacokinetics or toxic responses [67,68]. Such indirect involvement may affect the drug response to conditions such as fatty liver diseases, diabetic diseases, and inflammatory diseases.

## 5. The CYP4 Family and Inflammation

CYP4 enzymes are involved in inflammation through the metabolism of inflammatory molecules. They metabolize inflammatory mediators such as leukotrienes (LTs) and also produce 20-HETE [53]. While CYP4F11 possesses lower affinity toward leukotriene B4 (LTB4), neutrophilic CYP4F3A has the highest affinity for LTB4 ω-hydroxylation [36]. CYP4F3A metabolizes LTB4 into the inactive form 20-hydroxy leukotriene B4, mediating a critical step in regulation of the inflammatory response. However, CYP4A11 has shown low activity toward LTB4 using in vitro methods [69]. CYP4F3B ω-hydroxylates omega-3 eicosapentaenoic acid (EPA) and docosahexaenoic acid (DHA) to their 20-hydroxy and 22-hydroxy metabolites, respectively [32], which are lipid mediators that can activate inflammatory PPARs [70]. Studies have shown that hepatic and renal rat *CYP4F* genes were upregulated under inflammatory conditions following treatment with barium sulfate [71]. On the other hand, rat hepatic CYP4A mRNAs were downregulated in response to lipopolysaccharides used as a model of inflammation [72]. Human CYP4V2 was first identified in inflammatory cell macrophages, and its gene expression was reduced following selective treatment with a PPARγ agonist [41]. Depending on the clinical situation, ω-hydroxylase activity associated with the CYP4 family could be considered as a potential drug target for reducing the inflammatory response, providing a novel mechanism for future anti-inflammatory drugs.

## 6. The CYP4 Family and Cancers

Induction of CYP4 family members, including CYP4F2, CYP4F3, CYP4A11, and CYP4Z1, has been reported in various types of cancer [73,74]. Upregulation of CYPF2 and CYP4A11 was confirmed through Western blot assays in human thyroid, ovarian, breast, pancreatic, and colon cancer tissues [75]. CYP4Z1 is expressed in mammary tissue and upregulated in breast cancer tissue [74]. These findings suggest that ω-hydroxylase activity may be a biomarker of cancer prognosis. Evaluation of the CYP4 expression profile in hepatocellular carcinoma (HCC) showed that CYP4F2, CYP4F12, and CYP4V2 mRNA levels were negatively correlated with cell-cycle-associated genes, suggesting that these *CYP4* genes are favorable prognostic factors in HCC [76]. In addition, expression of CYP4 has been reported to be associated with angiogenesis through production of 20-HETE, which activates vascular endothelial receptors in arteries and thus increases blood supply to cancer cells [77]. Among CYP4 enzymes, CYP4F3B, CYP4A11, and CYP4F2 are major enzymes involved in the generation of 20-HETE, which plays an important role in tumor progression and angiogenesis. Therefore, their tissue expression and omega-hydroxylase activity levels play roles in cancer progression. CYP4B1 metabolizes several protoxic xenobiotics, including 2-aminofluorine, 2-naphthylamine, 4-ipomeanol, and benzidine [78,79,80,81]. Therefore, CYP4B1 involvement in cancers has been suggested based on its expression levels and metabolism of pro-carcinogens in the bladder and lung [78,82]. CYP4B1 may play a role in detoxification or activation in tissues. Sasaki et al. reported that the individuals carrying the *CYP4B1*2* allele have an increased risk of bladder cancer [27]. However, it has also been reported that there is no association between the *CYP4B1* genotype and the risk of lung cancer in the Japanese population [83]. Downregulation of CYP4B1 proteins represented an unfavorable indicator in patients with urothelial carcinomas of the upper urinary tract and bladder, indicating a protective role of CYP4B1 in patients with urotherial carcinomas [84]. Involvement of CYP4Z1 in breast cancer has been suggested, as it was identified in breast tissue and upregulated in breast carcinoma [74,76]. Therefore, CYP4Z1 was proposed as a biomarker for malignancy and/or progression of ovarian and prostate cancer [85]. It was reported that breast cancer cells exhibited the abnormal translocation of CYP4Z1 protein to the plasma membrane instead of targeting to the intracellular membrane of the endoplasmic reticulum, which caused the CYP4Z1 autoantibody production that might serve as a biomarker for the diagnosis [86]. Expression of CYP4Z1 has been reported to promote angiogenesis and tumor growth by increasing 20-HETE synthesis [74]. However, a recent functional study of CYP4Z1 in a recombinant enzyme system indicated that 20-HETE was not detected in the CYP4Z1 reaction with arachidonic acid, and suggested that CYP4Z1 may modulate breast cancer without direct 20-HETE synthesis [87]. Further studies are needed to clarify the roles of CYP4Z1 in carcinogenesis in various tissues.

## 7. The CYP4 Family and Cardiovascular Diseases

Several studies have shown that *CYP4* family genes are associated with cardiovascular diseases, including hypertension and myocardial infarction, through the production of 20-HETE or perturbation of fatty acid metabolism [88,89]. Multiple aspects of the mechanism underlying the effect of 20-HETE on the cardiovascular complex have been reported. In a metabolomics study in mice, increased 20-HETE levels in the blood (>120-fold) with chronic rofecoxib treatment were associated with reduced bleeding time and increased platelet aggregation [47]. Additionally, 20-HETE has been suggested to mediate androgen-induced hypertension through increasing the level of Cyp4a12 in a mouse study [90], wherein the increased level of Cyp4a12 produced more eicosanoids, which were predicted to mediate androgen-induced hypertension. In the kidney, however, 20-HETE exerts anti-hypertensive effects through inhibition of sodium reabsorption in the proximal tubule and thick ascending limb of Henle [91]. Furthermore, 20-HETE was found to act as a vasoconstrictor of vascular smooth muscle cells by allowing increased calcium entry into cells and enhanced phosphorylation of contractile elements [92,93,94]. Several studies have suggested interplay between 20-HETE and the renin–angiotensin aldosterone system (RAAS) in hypertension. Briefly, angiotensinogen II has been reported to increase renal production of 20-HETE [95], and 20-HETE can activate the RAAS by inducing angiotensin-converting enzyme [96,97]. Further investigations are needed to fully elucidate the mechanistic link between 20-HETE and the RAAS in humans. Rat CYP4A was downregulated in the kidney of hypertensive rats, which was associated with reduced formation of 20-HETE in the kidney and reduction of the diuretic effect [98]. CYP4A was upregulated in studies of doxorubicin-induced cardiotoxicity, where it was associated with myocardial infarction and increased 20-HETE synthesis [51]. Furthermore, Jarrar et al. found that heart cyp4a12 was highly upregulated in mice after cardiac toxicity induced by non-steroidal anti-inflammatory drugs [44]. Thus, targeting of 20-HETE synthesis or modulation of eicosanoid levels through manipulation of CYP4 enzymes can decrease the cardiotoxicity of such drugs. This application should be considered in future development of the drug for cardiovascular health care.

## 8. Role of the CYP4 Family in Other Diseases

Bietti’s crystalline dystrophy (BCD) is an autosomal recessive disease characterized by the presence of numerous small, yellow or white crystal-like deposits of fatty compounds in the light-sensitive retina tissue [52,99,100]. These deposits damage the retina, resulting in progressive atrophy of the retinal pigment epithelium and progressive vision loss at approximately 40 or 50 years of age [101,102]. The occurrence of BCD is more common in East Asian populations than other ethnic groups [103,104]. BCD is caused by mutations in the *CYP4V2* gene, which is comprised of 11 exons encoding a 525 amino acid protein on chromosome 4 [99,105,106]. CYP4V2 is known to metabolize fatty acids, and thus CYP4V2 in the retina is most likely involved in the breakdown and elimination of fatty acids from the retina [52]. Impaired CYP4V2 function due to genetic mutations may affect lipid metabolism and elimination from the retina. The severity and progression of BCD symptoms varies widely among patients. These variations may be influenced by differing levels of defectiveness in CYP4V2 function caused by mutations of different severities. Various mutations in *CYP4V2* have been found, including stop codon creation, an amino acid change in an important region, destruction of a splice site, and a frameshift in the CYP4V2 protein-coding cDNA. More than 60 mutations of the *CYP4V2* gene have been reported in BCD patients [99,103,105,107,108,109,110,111,112]. A number of mutations of *CYP4V2* have significant impacts on CYP4V2 activity. The most common mutation in BCD is an insertion–deletion mutation at the end of intron 6 and the beginning of exon 7 (IVS6-8del17insGC, c.802-8del17/insGC) [103,105,106,108,109,111,112,113,114,115,116,117,118,119,120,121,122,123,124,125]. This mutation causes the deletion of exon 7 in the CYP4V2 protein, resulting in a major structural change and the complete loss of CYP4V2 activity.

Type 3 lamellar ichthyosis, a skin keratinization disease, was found to be caused by genetic mutation of *CYP4F22* [126]. Since the discovery that *CYP4F22* is one of the causative genes for ichthyosis, the molecular mechanisms underlying the role of CYP4F22 in the etiology of ichthyosis have remained largely unknown until recently. Acylceramide is an important lipid of the skin permeability barrier, and patients with ichthyosis show strongly repressed acylceramide production [127,128,129,130]. Ohno et al. (2015) reported that CYP4F22 is responsible for the generation of acylceramide through ω-hydroxylation of long-chain fatty acids [18]. Recently, a *CYP4F22* genetic variant associated with lamellar ichthyosis was reported in a Tunisian family [131]. A missense mutation in exon 8, CYP4F22 Arg243Leu, was suggested to be linked to lamellar ichthyosis and predicted to be a functionally defective variant based on in silico analysis. Genetic screening for *CYP4F22* mutations associated with lamellar ichthyosis should be extended in future works.

## 9. Genetic Variants of the *CYP4* Family

### 9.1. Genetic Variants of CYP4B1

The first screening study for genetic polymorphism of *CYP4B1* was performed in French Caucasians and identified the new *CYP4B1* alleles *CYP4B1*2*, **3*, **4*, and **5* based on the P450 Nomenclature Committee [132]. Among them, *CYP4B1*2* caused a frameshift and premature stop codon, resulting in complete loss of CYP4B1 function. Two more alleles with frequencies <1%, *CYP4B1*6* and *CYP4B1*7*, were identified using a denaturing high-performance liquid chromatography method for 192 Japanese individuals [133]. Since CYP4B1 is involved in the metabolism of pro-carcinogens, its association with bladder cancer was investigated in a Japanese population, and subjects carrying the *CYP4B1*1/*2* or *CYP4B1*2/*2* genotypes exhibited a 1.75-fold increased risk of bladder cancer [27]. This finding might be explained as the loss of function allele *CYP4B1*2* providing lower capacity for activation of carcinogenic compounds. However, a lung cancer risk study of *CYP4B1*1–*7* showed no association with lung cancer in a Japanese population [83]. Further studies are needed to determine its association with lung cancer using a large cohort. Study of structure–function relationships has been essential to understanding the efficiency of catalytic activity as well as to explaining the varying degrees of molecular defectiveness of the protein mutants. Investigation of local peptide structures on the CYP4B1 protein and their roles in heme stability with catalytic function has been reported [134,135,136], and these data will be important to understand inter-individual variations in the activity of CYP4B1 coding variants.

### 9.2. Genetic Variants of CYP4A11, CYP4F2, 4F11, and CYP4F22

Among *CYP4* family genes, *CYP4A11* and *CYP4F2* have been extensively studied in association with warfarin dosage and the cardiovascular complex. Genetic variants of *CYP4F2* and *CYP4A11* genes are reportedly associated with cardiovascular diseases such as hypertension [137,138,139]. More than 3400 single nucleotide polymorphisms (SNPs) of human *CYP4A11* and 5900 SNPs of the *CYP4F2* gene have been reported in the NCBI database to date. However, only a small number of the SNPs have been shown to have clinical associations with functional changes. One of the most extensively studied SNPs of *CYP4A11* is a variant of rs1126742 that causes an amino acid change of Phe434 to Ser, leading to reduced 20-HETE synthesis from arachidonic acid [140,141]. Since the discovery of the functional role of CYP4A11 in the synthesis of 20-HETE, the association of *CYP4A11* polymorphisms with cardiovascular risk has been studied extensively in humans [142,143,144,145,146,147,148]. The US Food and Drug Administration recommends genotyping of *CYP4F2* variants for determination of warfarin doses [149,150]. The *CYP4F2* genetic variant rs2108622 is a non-synonymous variant that causes a change in the amino acid sequence of valine to methionine and exhibits reduced enzymatic activity toward the metabolism of vitamin K [62]. Since individuals with reduced activity of CYP4F2 for vitamin K inactivation may have higher levels of warfarin than individuals with *CYP4F2*1/*1*, higher maintenance dosages of warfarin have been recommended for individuals with reduced *CYP4F2* alleles [149]. Many studies have attempted to develop an accurate warfarin dosing algorithm using multiple genes, such as *CYP2C9*, *VKORC1*, and *CYP4F2* [151,152,153,154,155]. Studies regarding *CYP4A22* genetic polymorphisms have been limited to certain populations, such as Japanese and French populations [25,156]. The association of *CYP4A22* variants with human diseases has still not been investigated, which might be due to low expression levels of the *CYP4A22* gene. The *CYP4F3* gene undergoes alternative splicing to form the CYP4F3A and CYP4F3B enzymes, depending on the cell type [157]. Genome-wide investigation showed that the functional SNP *CYP4F3* rs4646904 was associated with lung cancer, especially in smokers [30]. However, the functionality of this SNP in lung cancer pathology remains unidentified. In addition, a high intake of polyunsaturated fatty acids was associated with reduced risk of ulcerative colitis in patients with *CYP4F3* rs4646904 GG/AG, but not those with the AA genotype [158]. Regarding the *CYP4F11* gene, Yi et al. found through in vitro methods that CYP4F11 D315N protein showed approximately 50% and 32% decreases in intrinsic clearance of erythromycin and arachidonic acid, respectively, compared to the wild type [37]. The *CYP4F11* variant (rs1060463) was associated with small bowel bleeding risk induced by aspirin [159]. Seven variants with amino acid changes in the *CYP4F12* gene were identified and functional changes were investigated using ebastine as a substrate [160]. In their report, two coding variants, Val90Ile and Arg188Cys, exhibited significantly decreased activity toward ebastine hydroxylation. The intronic variant *CYP4F12* rs11085971, which contains a nucleotide substitution of guanine to thymine, was identified as a candidate oxidative-stress-related genetic marker for the development of type 1 lesions in cerebral cavernous malformation, and could serve as an early objective predictor of disease outcome [161]. Since the discovery of *CYP4F22* was linked to its association with lamellar ichthyosis [18], genetic studies of *CYP4F22* polymorphisms have been undertaken. A *CYP4F22* variant, CYP4F22 Arg243Leu, was associated with lamellar ichthyosis in a Tunisian family [131], and further genetic studies should be conducted in clinical settings.

### 9.3. Genetic Variants of Other CYP4 Genes

Genetic polymorphism studies of *CYP4V2* with respect to BCD are described above. In addition to BCD, genome-wide analysis found that a *CYP4V2* genetic variant was strongly associated with deep vein thrombosis [162], which was confirmed later in multiple studies [163,164]. Association of the genetic variant 7234C>A (rs13146272) on exon 6 of the *CYP4V2* gene with the risk of deep venous thrombosis and tamoxifen-induced venous thrombosis has been reported [165]. The exact mechanism through which the *CYP4V2* gene defect increases the risk of deep vein thrombosis remains poorly understood. This genetic variant substitutes polar glutamine with basic lysine at position 259 of the CYP4V2 amino acid sequence, which may influence its activity. Genetic studies of *4Z1* and *4X1* are scarce, as these genes were recently identified and their physiological roles remain unclear. CYP4X1 was found to convert the endocannabinoid anandamide, an important signaling molecule in the neurovascular cascade, into a single monooxygenated product (14,15-epoxyeicosatrienoic ethanolamide), suggesting a potential role in brain signaling [40]. High levels of mRNA expression of CYP4X1 were found in the skin, brain, heart, liver, prostate, and breast [40] and CYP4Z1 mRNA was preferentially expressed in mammary tissue [21]. Functional analysis of CYP4Z1 indicated that CYP4Z1 has catalytic activity toward lauric and arachidonic acids, but 20-HETE was not detected in arachidonic acid metabolism [87]. Major genetic polymorphisms in *CYP4* genes with clinical impact were summarized in Table 2. 

## 10. Linkage Disequilibrium among *CYP4* Genes

Five *CYP4* genes, *CYP4A22, CYP4A11, CYP4B1, CYP4X1,* and *CYP4Z1*, are located on chromosome 1 [4]. A number of studies based on next generation sequencing tools and a 1000-genome project have identified SNPs in these genes. However, their functional roles, clinical relationships, and linkage disequilibrium (LD) are poorly characterized. From the 1000-genome database, a total of 14 coding SNPs with > 5% global minimum allele frequency were identified for *CYP4A22, CYP4A11*, and *CYP4B1*, and this LD block was analyzed (Figure 1A). Ethnically distinct populations exhibited differing LD blocks and haplotype structures. No strong LD was found among these three *CYP4* genes that are clustered on chromosome 1. Six *CYP4F* genes, including *CYP4F2, CYP4F3, CYP4F8, CYP5F11, CYP4F12*, and *CYP4F22*, are located on the same chromatid of chromosome 19 [4]. Using the same method, coding variants with > 5% global frequency were selected from a 1000-genome database and their haplotypes and LD were analyzed (Figure 1B). As illustrated in Figure 1A, ethnically distinct groups showed differing frequencies and LD structures. An LD block covering more than one *CYP4* gene was not observed for *CYP4F* genes in coding variant analysis. Instead, a strong linkage was found between *CYP4F2* (rs2074900) and *CYP4F11* (rs8104361) in a Western European population. Since *CYP4* genes on the same chromosome with highly similar DNA structures can act as a linkage unit or as independent genes, further linkage analysis using more validated SNPs over all regions of *CYP4* genes is needed to improve the current knowledge of *CYP4* genetics.

## 11. Conclusions and Future Prospects

CYP4 enzymes are responsible for the metabolism of fatty acids and play important roles in the homeostasis of fatty acids and fatty-acid-derived biomolecules such as leukotriene, prostanoid, and 20-HETE. Thus, CYP4 enzymes make important contributions to human health, including cardiovascular health, skin barrier maintenance, eye function, and cancer protection. However, the lack of research into certain aspects of the CYP4 family must be overcome. First, a specific antibody for the detection of each CYP4 protein and a specific substrate for each enzyme function must be developed to clearly determine the expression levels of these enzymes in different tissues under various induction, inhibition, and genetic conditions. High similarity of protein structures, overlapping substrates, co-expression in the same tissues, and genetic differences among individuals have interfered with the identification and characterization of *CYP4* genes. For targeted therapy and targeted delivery of drugs into cells or specific tissues, accurate measurement of CYP4 activity in tissues is essential. Second, further functional studies of *CYP4* genetic variants are needed. A growing number of genetic mutations of *CYP4* genes have been identified using high-throughput sequencing techniques. However, most of their functional changes compared to the wild type remain unknown. Only a small number of high-frequency genetic variants with known functional information have been investigated in multiple populations, likely due to their high statistical power, which enables publication. Although in silico tools are useful for the prediction of functional changes, in silico prediction does not yet perfectly reflect in vivo conditions. Therefore, various commercial software programs often provide inconsistent predictions for the same genetic mutations. Development of high-throughput techniques for in vitro functional study and improvement of in silico methods are needed to elucidate the functional changes caused by mutations. Third, globally standardized values for CYP4 activity must be developed for application in artificial intelligence technology and algorithms used for the prediction of CYP4-related human diseases or the progression of disease states. As shown in Figure 2, large variations in CYP4-mediated metabolism, genetic variants of *CYP4* and other genes, and differing environmental conditions have been observed among individuals. Data integration to support correct diagnosis in humans is currently not possible, but is the ultimate goal of such research. To achieve this goal, accurate molecular tools for characterization of each CYP4 enzyme, functional information about *CYP4* genetic variants, and a standardized system for the application of CYP4 functional values in artificial intelligence or machine-learning tools are needed for personalized health care.

## Figures and Tables

**Figure 1 ijms-20-04274-f001:**
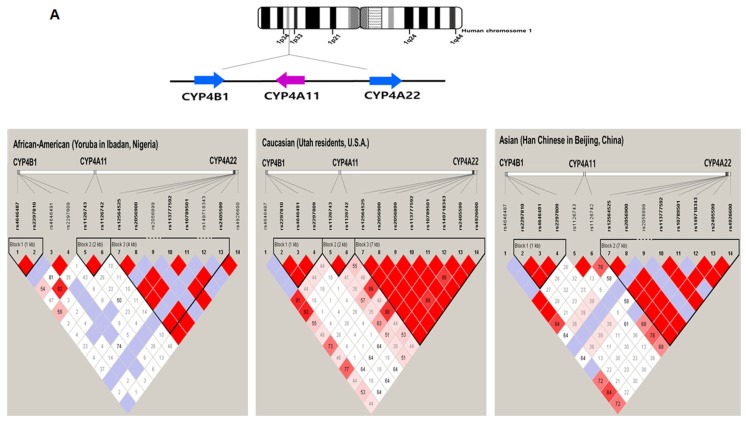
Linkage disequilibrium (LD) plots of *CYP4* genetic variants in African, Caucasian, and Asian populations. Populations in Yoruba, Utah, and Beijing represent African, Caucasian, and Asian populations, respectively. The coding single nucleotide polymorphisms (SNPs) with a minor allele frequency (MAF) of 0.05 or greater in the 1000 genome data base were selected to avoid estimation errors in linkage analysis. (**A**) LD structures of *CYP4A11*, *CYP4A22*, and *CYP4B1* with common coding SNPs. *CYP4A11*, *CYP4A22*, and *CYP4B1* are clustered on chromosome 1. The SNPs, shown from left to right within the figure, are as follows: rs4646487, rs2297810, rs4646491, rs2297809, rs1126743, rs1126742, rs12564525, rs2056900, rs2056899, rs113777592, rs10789501, rs149718343, rs2405599, and rs4926600. (**B**) LD structures of *CYP4F2*, *CYP4F3*, *CYP4F11*, and *CYP4F12* using common coding SNPs. *CYP4F2*, *CYP4F3*, *CYP4F11*, and *CYP4F12* are clustered on chromosome 19. The SNPs, shown from left to right within the figure, are as follows: rs1805040, rs7254013, rs16995376, rs16995378, rs609636, rs609290, rs2285888, rs593818, rs3093200, rs2108622, rs2074900, rs3093105, rs1060463, and rs8104361. The numbers in squares refer to pairwise LD values, measured as D’ (coefficient of linkage disequilibrium). Red depicts a significant linkage between a pair of SNPs. Numbers inside squares indicate the D’ value multiplied by 100.

**Figure 2 ijms-20-04274-f002:**
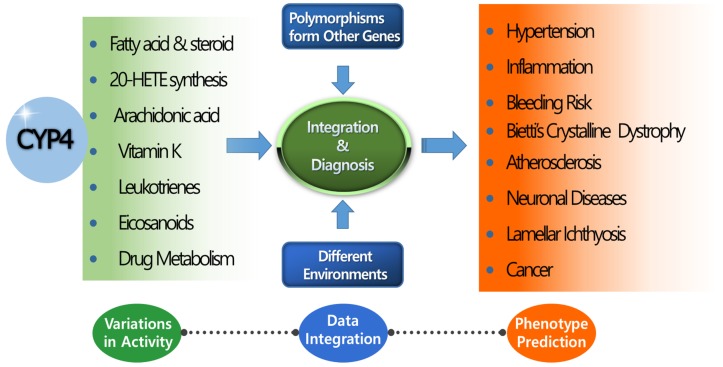
Correlation of *CYP4* genes with phenotypic outcomes. Most *CYP4* genes share similar structures and overlapping metabolic substrates. Phenotypic outcome prediction is difficult with a single or few *CYP4* genetic studies. Phenotypic outcomes are affected by genetic polymorphisms of various genes and dynamic environmental factors. Fundamental research into *CYP4* genes is essential to provide the data integration necessary for more accurate phenotype prediction than can be obtained using conventional methods.

**Table 1 ijms-20-04274-t001:** Substrates and major expression tissues of cytochrome P450 4 (CYP4) enzymes in humans.

CYP4 Enzyme	Expression Tissues	Substrates	Reference
***CYP4A11***	Platelets, liver, kidney	Lauric acid, myristic acid, arachidonic acid	[6,23,24]
***CYP4A22***	Low level in liver	Arachidonic acid	[25,26]
***CYP4B1***	Lung, bladder, fat tissues	2-aminofluorene, 2-naphthylamine, benzidine, arachidonic acid	[27,28]
***CYP4F2***	Liver, platelet, kidney	Arachidonic acid, lauric acid, vitamin K, leukotriene	[6,29]
***CYP4F3A***	Neutrophils, monocytes, eosinophils	Leukotriene B4	[12,30,31]
***CYP4F3B***	Liver, kidney, trachea, gastro intestinal tract	Eicosapentaenoic acid, arachidonic acid	[12,32]
***CYP4F8***	Prostate, seminal vesicles, epidermis, hair follicles, sweat glands, corneal epithelium, proximal renal tubules, epithelial linings of gut and urinary tract	Arachidonic acid, prostaglandin H, prostaglandin E2	[33,34,35]
***CYP4F11***	Liver, kidney, heart, skeletal muscle, gall bladder, keratinocytes	Vitamin K, erythromycin, arachidonic acid	[11,29,36,37,38]
***CYP4F12***	Liver, kidney, colon, small intestine, heart, eosinophils, neutrophils	Arachidonic acid, leukotriene B4, ebastine	[8,31,33,39]
***CYP4F22***	Skin	Ultra-long-chain fatty acid (acylceramide production)	[18]
***CYP4X1***	Skin, breast, brain, heart, liver, prostate, trachea, aorta	Anandamide, arachidonic acid	[21,40]
***CYP4V2***	Macrophages, retina cells, cornea cells	Arachidonic acid, lauric acid, eicosapentanoic acid, docosahexanoic acid	[19,41]
***CYP4Z1***	Mammary tissues, cancer cells	Lauric acid, myristic acid	[21,42]

**Table 2 ijms-20-04274-t002:** Representatives of genetic polymorphisms in *CYP4* genes with clinical impact and their frequencies in different ethnic groups.

Gene	SNP	Location	Mutation	Effect	Frequency ^a^			Functional Effect
					European	African	Asian	
*CYP4A11*	rs1126742	Exon	A > G	Phe330Ser	0.15	0.36	0.25	It was associated with hypertension in white individuals, most probably through decreased production of 20-HETE in the kidney [137].
*CYP4F2*	rs2108622	Exon	C > T	Val433Met	0.27	0.06	0.26	It reduced the metabolism of vitamin K. Therefore, patients carrying this genetic variant needed a higher dose of warfarin, in order to keep the targeted anticoagulant effect [149,150,151].
*CYP4F3*	rs4646904	Exon	A > G	Val358Val	0.65	0.35	0.34	It was associated with lung cancer, especially in smokers [30] and ulcerative colitis [158].
*CYP4F11*	rs200033002	Exon	C > T	Asp315Asn	0	0	0.01	It decreased the metabolism of erythromycin and arachidonic acid compared to the wild type in vitro [37].
*CYP4B1*	rs3215983	Frameshift variant	AT881–882del	Produces premature stop codon	0.15	ND	0.33	It was reported to increase the risk of bladder cancer, because it has lower capacity to metabolize the carcinogenic compounds [27].
*CYP4V2*	rs13146272	Exon	C > A	Gln259Lys	0.36	0.4	0.6	It was associated with the risk of deep venous thrombosis and tamoxifen-induced venous thrombosis [162,163,164,165].
*CYP4V2*	rs199476197	Exon	A > C	His331Pro	0	0	0.0004	It decreased CYP4V2 protein expression and activity toward fatty acid metabolism. Therefore, this genetic variant may cause accumulation of fatty acids in the retina [19,166].
*CYP4V2*	IVS6-8del17insGC	Intron 6, exon 7	Insertion/deletion	Exon7 del	ND	ND	ND	It causes deletion of exon 7 in the CYP4V2 protein, resulting in a complete loss of CYP4V2 activity. It is the most common mutations in BCD patients [52,99].

^a^ Data regarding the frequency of genetic variants among different ethnic groups were obtained from Ensemble database. ND, not determined. BCD, Bietti’s crystalline dystrophy.

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
