# Peer review of "Molecular Functionality of Cytochrome P450 4 (CYP4) Genetic Polymorphisms and Their Clinical Implications"

_ijms, 2019, doi:10.3390/ijms20174274_

Round 1
Reviewer 1 Report
These authors have reviewed the functionality of the cytochrome P450 4 class of liver enzymes. This is a comprehensive review. Comments:
The group 4 isoenzymes are different from the more widely studied 1-3 class because of its greater role in genetic disease risk whereas the others are more closely linked to pharmacogenomic risk. This is an important distinction that requires more clarification. While it is accurate to call these cytochrome P450 variants, this reviewer wonders if there is another name that can be given to this family of hepatic isoenzymes so that this confusion does not perpetuate?
The authors do describe pharmacogenomic implications of CYP4 variants such as its efffect on the pharmacokinetics of warfarin. Are any variants linked to any drug's pharmacodynamic effect?
Defects in fatty acid metabolism causes significant inborn errors of diseases that are routinely screened for using dried blood spots. How does the P450 variants correlate to variants in the other enzymes that metabolize long and median chain fatty acids such as the dehydrogenases, transferases, and translocases?
Author Response
Reviewer 1.
Comments and Suggestions for Authors
These authors have reviewed the functionality of the cytochrome P450 4 class of liver enzymes. This is a comprehensive review. Comments:
The group 4 isoenzymes are different from the more widely studied 1-3 class because of its greater role in genetic disease risk whereas the others are more closely linked to pharmacogenomic risk. This is an important distinction that requires more clarification. While it is accurate to call these cytochrome P450 variants, this reviewer wonders if there is another name that can be given to this family of hepatic isoenzymes so that this confusion does not perpetuate?
We agree to the reviewer’s comment. According to the reviewer suggestion, the following sentence was added in the introduction section in line 35-38 as follows:
Generally, CYP families 1, 2, and 3 include major xenobiotic-metabolizing enzymes responsible for their major roles in pharmacogenomic risk, while CYP4 enzymes are involved in the metabolism of fatty acids with their close links to genetic disease risk.
As far as I know, cytochrome P450 4 subfamily enzymes or CYP4 enzymes are generally called, instead of isozymes. There are no other for P450s. If someone want to use hepatic isozymes for P450s, the use of isoenzymes should be used together with CYP family or CYP subfamily, for example CYP4F isoenzymes.
The authors do describe pharmacogenomic implications of CYP4 variants such as its efffect on the pharmacokinetics of warfarin. Are any variants linked to any drug's pharmacodynamic effect?
CYP4F2 affected the pharmacokinetics of vitamin K1, which is the coagulation factor and used as antidote of warfarin overdose. The area under the curve of vitamin K1 was significantly higher among CYP4F2*3 than CYP4F1 carriers. In addition, CYP4F2 genetic variants affect the pharmacodynamics of warfarin which inhibits activation of vitamin K. CYP4F2 genetic variant with pharmacokinetics and pharmacodynamics effects of warfarin was described in the Table 2 with the reference papers.
Defects in fatty acid metabolism causes significant inborn errors of diseases that are routinely screened for using dried blood spots. How does the P450 variants correlate to variants in the other enzymes that metabolize long and median chain fatty acids such as the dehydrogenases, transferases, and translocases?
From our literature search, there have been no studies regarding the correlation of genetic variants of CYP4 genes with dehydrogenases, transferases, and translocases genetic variants. As reviewer’s suggestions, it would be great if we study the combined analysis of CYP4 and those enzymes.
Reviewer 2 Report
This review focuses on the CYP4 family of enzymes in terms of their functional roles, genetic variations, and influences on human diseases.The manuscript is interesting and the idea is nice. The title is clear and it is adequate to the content of the article. The conclusions or summary are accurate and supported by the content. The author’s work on discussing achieved results is appreciated. The minor revisions are necessary to improve clarity of the presentation. I have some recomandations for authors:
- Moderate English changes required;
- Please extend the references with recent literature data. In this sense cite this: https://link.springer.com/article/10.1007/s00226-018-1071-5;
- Please improve the quality of figure 2.
Author Response
Reviewer 2.
Comments and Suggestions for Authors
This review focuses on the CYP4 family of enzymes in terms of their functional roles, genetic variations, and influences on human diseases.The manuscript is interesting and the idea is nice. The title is clear and it is adequate to the content of the article. The conclusions or summary are accurate and supported by the content. The author’s work on discussing achieved results is appreciated. The minor revisions are necessary to improve clarity of the presentation. I have some recomandations for authors:
- Moderate English changes required;
Thank you very much for the review comment. The English in this document has been professionally checked by two professional editors, both native speakers of English. For a certificate, please see: http://www.textcheck.com/certificate/h1KUpK.
- Please extend the references with recent literature data. In this sense cite this: https://link.springer.com/article/10.1007/s00226-018-1071-5;
We tried several times, but the page is not available. Please check again.
- Please improve the quality of figure 2.
We improved the quality of figure 2 as attached.